# Differential Modulation of Matrix Metalloproteinases-2 and -7 in LAM/TSC Cells

**DOI:** 10.3390/biomedicines9121760

**Published:** 2021-11-24

**Authors:** Silvia Ancona, Emanuela Orpianesi, Clara Bernardelli, Eloisa Chiaramonte, Raffaella Chiaramonte, Silvia Terraneo, Fabiano Di Marco, Elena Lesma

**Affiliations:** 1Department of Health Sciences, Università degli Studi di Milano, 20142 Milan, Italy; silvia.ancona@unimi.it (S.A.); emanuelaorpianesi@gmail.com (E.O.); clara.bernardelli@unimi.it (C.B.); eloisa.chiaramonte@gmail.com (E.C.); raffaella.chiaramonte@unimi.it (R.C.); silvia.terraneo10@gmail.com (S.T.); fabiano.dimarco@unimi.it (F.D.M.); 2Respiratory Unit, Azienda Socio Sanitaria Territoriale Santi Paolo e Carlo, 20142 Milan, Italy; 3Respiratory Unit, Azienda Socio Sanitaria Territoriale-Papa Giovanni XXIII Hospital, 24127 Bergamo, Italy

**Keywords:** MMP-2, MMP-7, LAM, TSC, cell motility

## Abstract

Matrix metalloproteinase (MMP) dysregulation is implicated in several diseases, given their involvement in extracellular matrix degradation and cell motility. In lymphangioleiomyomatosis (LAM), a pulmonary rare disease, MMP-2 and MMP-9 have been detected at high levels in serum and urine. LAM cells, characterized by a mutation in the tuberous sclerosis complex (TSC)1 or TSC2, promote cystic lung destruction. The role of MMPs in invasive and destructive LAM cell capability has not yet been fully understood. We evaluated MMP-2 and MMP-7 expression, secretion, and activity in primary LAM/TSC cells that bear a TSC2 germline mutation and an epigenetic modification and depend on epidermal growth factor (EGF) for survival. 5-azacytidine restored tuberin expression with a reduction of MMP-2 and MMP-7 levels and inhibits motility, similarly to rapamycin and anti-EGFR antibody. Both drugs reduced MMP-2 and MMP-7 secretion and activity during wound healing and decreased their expression in lung nodules of a LAM mouse model. In LAM/TSC cells, MMP-2 and MMP-7 are dependent on tuberin expression, cellular adhesion, and migration. MMPs appears sensitive to rapamycin and anti-EGFR antibody only during cellular migration. Our data indicate a complex and differential modulation of MMP-2 and MMP-7 in LAM/TSC cells, likely critical for lung parenchyma remodeling during LAM progression.

## 1. Introduction

Matrix metalloproteinases (MMPs), a group of zinc-dependent endopeptidases, is the principal family of enzymes responsible for extracellular matrix (ECM) degradation [1]. This event plays a fundamental role in physiological processes, including organ development, tissue remodeling, angiogenesis, inflammation, and wound healing. Since the ECM degradation is crucial also for cellular migration, the proteolytic activity of MMPs is exploited by cancer cells to invade nearby tissue and reach the blood vessels to extravasate, ultimately promoting metastatic progression [2]. MMPs are either secreted from the cells or anchored to the plasma membrane [3] and their regulation involves both the control of mRNA levels and the cleavage of the zymogen to the active form. Moreover, protein secretion and localization are highly controlled, and their function in vivo strongly depends on the presence of specific inhibitors [2]. In the last decades, the transmembrane protein CD147 was indicated as a potent activator of MMPs [4]. The interest on this protein has grown because it was demonstrated that CD147 enhances the invasion and survival of cancer cells mediating the activity of MMP-1, MMP-2, MMP-3, MMP-9, and MMP-11. Moreover, the studies suggest a functional relationship of CD147-MMP-VEGF expression to induce tumor angiogenesis [4,5].

Imbalance between MMPs and their inhibitors is implicated in a variety of pulmonary disorders including lymphangioleiomyomatosis (LAM) [6,7]. LAM is a rare progressive interstitial lung disease affecting predominantly young women and leading to loss of pulmonary function, characterized by the destruction of lung parenchyma with the development of LAM nodules, thin-walled cysts, and deposits of spindle-shaped and epithelioid cells [8,9]. LAM cells have a benign histologic appearance, but genetic and cellular evidence has shown that they exhibit the features and behaviors of a neoplasm, driving to the reclassification of this disease as a low-grade, destructive, metastasizing neoplasm [10]. Smooth muscle-like LAM cells display loss of heterozygosity in the TSC (tuberous sclerosis complex) 1 or TSC2 tumor suppressor genes that codes for hamartin and tuberin, respectively, leading to hyperactivation of mTORC1 (mammalian target of rapamycin) [11]. For this reason, rapamycin has been approved for LAM treatment, but novel therapeutic targets must be identified to improve LAM therapy because it is highly debilitating for patients and hard to suspend, as lung function declines when the therapy is stopped [12]. In our previous studies, we characterized LAM/TSC cells isolated from the chylous thorax of a patient affected by LAM associated with TSC [13]. These cells bear a TSC2 germline mutation and do not express tuberin for an epigenetic modification. Indeed, the treatment with the chromatin-remodeling agent 5-azacytidine restores tuberin expression. The proliferation of LAM/TSC cells strongly depends on the epidermal growth factor (EGF), and blockade of EGF receptors with the anti-EGFR antibody causes cell death [13,14,15]. Interestingly, these cells can survive and proliferate both in adherent and non-adherent conditions and can migrate in vitro; for this reason, LAM/TSC cells were used to induce lung lesions in an innovative mouse model of LAM [16].

Recent studies demonstrated that LAM cells destroy the lung tissue by proteolytic activity through the secretion of MMPs [17]. The levels of MMP-1, MMP-2, MMP-9, and MMP-14 are increased both in the serum of the LAM patients and/or expressed in lung tissue adjacent to cysts [7,18,19]. Interestingly, we recently demonstrated that MMP-2 and MMP-7 serum levels might be used as biomarkers for LAM diagnosis [20]. Moreover, MMP-2 and MMP-9 might play a role in cystic lung destruction [21]. Indeed, MMP-2 is responsible for the breakdown of ECM components, such as type IV collagen, and high concentrations of its transcript have been found in cultured tuberin-null LAM cells derived from patients [22]. In addition, it was demonstrated that MMP-9 is one of the main immunohistochemical and histological parameter associated with loss of pulmonary function in LAM [23]. These observations provide a strong rationale to deepen MMP function in LAM.

In clinics, several MMP inhibitors were approved. Among those, doxycycline was used as a broad-spectrum MMP inhibitor, especially in periodontal diseases [24]. Doxycycline reduced the production of MMP-2 both in LAM cells and in TSC2-null mouse fibroblasts [25]. Moreover, the administration of doxycycline significantly reduced the levels of urinary MMP-9 and serum MMP-2 in LAM patients, even if there was association between MMPs blockade and improvement in lung function only in patients with a mild disease [26]. Moreover, the inefficacy of doxycycline in reducing lung decline in LAM patients has also been reported [27]. However, even only the investigation of MMPs in LAM gives way to the possibility that lung destruction might involve the metalloproteinase system and, at the same time, might depend also on other factors. For this reason, the American Thoracic Society and the Japanese Respiratory Society suggested that the clinical outcomes derived from the combination of doxycycline with mTOR inhibitors and hormonal therapy, which are already used in LAM patients, be deepened [28].

MMP-2 and MMP-9 are activated by MMP-7 [29]. MMP-7 is produced by the stromal cells surrounding cancer cells and by the cancer cells themselves, as in invasive tumor cells in which it is often overexpressed [30]. Indeed, MMP-7 has powerful proteolytic activity and broad substrate specificity, providing an important role in the invasion and metastasis of several carcinomas, including lung carcinomas [31]. Finally, MMP-7 mediates the invasion of rat tuberin-null cells and is expressed in LAM tissues, suggesting a role for this protein in the capability of tuberin-deficient cells to acquire the invasive properties that may underlie the development of LAM [32].

Here, we investigated the expression of MMP-2 and MMP-7 in human LAM/TSC cells and how their secretion contributes to cell motility. Non adherent LAM/TSC cells express higher levels of both MMP-2 and MMP-7, confirming the role of these proteins in sustaining the invasive capability of LAM/TSC cells through ECM degradation. LAM/TSC motility in a wound healing assay was significantly reduced by the treatment with rapamycin and the anti-EGFR antibody, which, interestingly, decreased also MMP-2 secretion during wound closure. As further confirmation of this result, the induction of tuberin expression in LAM/TSC cells reduced both MMP-2 expression and cell motility. In vivo, anti-EGFR antibody and rapamycin mainly reduced MMP-2 in lung nodules but not in the epithelium. On the contrary, MMP-7 expression was decreased both in nodules and in epithelial cells after anti-EGFR treatment, while rapamycin showed primarily a marked inhibition of MMP-7 in nodules. Taken together, these data suggest an important role of MMP-2 and -7 in lung parenchyma remodeling during LAM progression, making these proteins an important target to deepen in the communication between pathological cells and their microenvironments.

## 2. Materials and Methods

### 2.1. Cell Cultures and Treatments

LAM/TSC cells were isolated, characterized, and grown as previously described [13]. Cells were obtained from chylous thorax of a LAM/TSC2 patient who had given written informed consent according to the Declaration of Helsinki. The study was approved by the Institutional Review Board of Milan’s San Paolo Hospital. Culture medium used for LAM/TSC cells contained a 1:1 mixture of DMEM/Ham F12 (Euroclone, Milano, Italy) supplemented with 200 nmol/L hydrocortisone (Sigma-Aldrich, St. Louis, MO, USA), 10 ng/mL epidermal growth factor (EGF; Sigma-Aldrich, St. Louis, MO, USA), and 15% fetal bovine serum (Euroclone, Milano, Italy). As previously reported, these cells can be grown as a stabilized cell line and were routinely checked for morphological, biochemical, and genetic features.

MCF7 breast cancer cells (ATCC, Rockville, MD, USA) were maintained in DMEM supplemented with 10% FCS.

LAM/TSC cells were incubated with anti-EGFR antibody (5 µg/mL; Calbiochem, San Diego, CA, USA, Cat. No. GR13L), rapamycin (5 ng/mL; Rapamune-Sirolimus, Wyeth Europa, Sandwich, Kent, UK, Lot. No. 19727), 5-azacytidine (1 µM, Sigma-Aldrich, St. Louis, MO, USA, Cat. No. A2385), or doxycycline hydrochloride (0.5 µM and 5 µM; Sigma-Aldrich, St. Louis, MO, USA, Cat. No. D3447) at time points as indicated in each experiment.

### 2.2. Western Blot Analysis

Cells were lysed in lysis buffer (5 mM EDTA, 100 mM deoxycholic acid, 3% sodium dodecyl sulfate, Sigma-Aldrich). Samples (25 µg per lane) were boiled for 5 min and analyzed by 10% sodium dodecyl sulfate–polyacrylamide gel electrophoresis. After transfer to nitrocellulose membranes (Amersham, Arlington Height, IL, USA) and blocking at room temperature for 1 h with 5% dry milk (Merck, Darmstadt, Germany), membranes were incubated overnight at 4 °C with antibodies against MMP-2 (1:1000, Cell Signaling Technologies, Danvers, MA, USA, Cat. No. 40994), MMP-7 (1:1000; Cell Signaling Technologies, Cat. No. 71031), tuberin (1:1000; Cell Signaling Technologies, Cat. No. 3635), phospho-S6 (1:1000; Cell Signaling Technologies, Cat. No. 2211), or β-actin (1:1000; Sigma-Aldrich, Cat. No. A5441). Membranes were washed and incubated for 1 h with the appropriate secondary horseradish peroxidase conjugate antibodies (1:10,000; Thermo Scientific, Rockford, IL, USA, Cat. No. 31430/31460). The reaction was revealed by using the ECLT Prime Western Blotting System (Amersham). Images were acquired on a Kodak image station 440 CF, and densitometric analysis was performed by using Kodak 1D 3.6 Software Image (Kodak, Milano, Italy). Results were expressed as the mean value from at least three independent experiments relative to β-actin levels.

### 2.3. ELISA

The total amount of MMP-2 and MMP-7 protein in the media was measured using the enzyme-linked immunosorbent assay (ELISA) kit (Quantikine Human MMP-2 or MMP-7 ELISA kit) according to the manufacturer’s instruction (R&D Systems, Minneapolis, MN, USA).

### 2.4. Wound Healing Assay

Confluent monolayers of LAM/TSC cells treated with anti-EGFR antibody, rapamycin, 5-azacytidine, and doxycycline, as above described, were scraped with a 200 µL pipette tip. Culture medium was changed to remove detached and damaged cells, and wound closure was monitored microscopically at different time points (9, 11, 19, 25, and 35 h). LAM/TSC cells were wounded at the same time in three independent experiments, and migration was determined using the ImageJ program as an average closed area of the wound relative to the initial wound area.

### 2.5. Flow Cytometric Analysis

LAM/TSC cells were collected by centrifugation and washed in PBS. Cells were fixed with fixation buffer (BD, Becton Dickinson, Milano, Italy) for 1 h at 4 °C. Cells were washed and permeabilized with Perm/Wash buffer I (BD, Becton Dickinson, Milano, Italy) for 1 h at RT. Permeabilized samples were incubated with anti-human CD147 PE-conjugated antibody (Immunotools, Friesoythe, Germany) for 1 h at RT and then with mouse IgG1 isotype control PE-conjugated (1:100; Immunotools, Friesoythe, Germany, Cat. No. 21275534S). After washing twice with Perm/Wash buffer, samples were analyzed by Cytomics FC500 (Beckman Coulter, Brea, CA, USA). Data acquisition and analysis were done using CXP 2.2 software (Beckman Coulter, Brea, CA, USA).

### 2.6. Animal Experiments and Pharmacological Treatments

A mouse model was obtained as previously described in Lesma et al. [16]. All experimental procedures were performed in accordance with the Italian Guidelines for Laboratory Animals, which conforms to the European Committees Directive (86/609/EEC), and study protocols were previously approved by the Ministero della Sanità (2/2011 Protocol).

At 26 weeks after cells administration, mice were randomly divided into four groups: (a) LAM/TSC mice, (b) LAM/TSC mice treated with anti-EGFR antibody, (c) LAM/TSC mice treated with rapamycin, and (d) control mice treated with the vehicle. Anti-EGFR antibody was administrated intraperitoneally (i.p.) twice weekly at a starting dose of 400 mg/m^2^ followed by a subsequent dose of 250 mg/m^2^ (Merck, Darmstadt, Germany), and 4 mg/kg rapamycin (Rapamune-Sirolimus; Wyeth Europa, Sandwich, Kent, UK) twice weekly for 4 weeks. At 30 weeks after cell administration, the mice were sacrificed, and blood, lungs, and lymph nodes removed. All specimens were fixed in 4% paraformaldehyde at 4 °C overnight and embedded in paraffin.

### 2.7. Immunohistochemical Analysis

Immunohistochemistry was carried out on de-paraffinized, rehydrated 4 μm tissue sections. For antigen retrieval, the slides were treated under pressure at 95 °C in citrate buffer, pH 6.0 for 5 min and, after cooling, incubated with 0.3% H_2_O_2_ in methanol for 20 min to quench endogenous peroxidase activity. Sections were incubated with 3% albumin bovine serum (BSA) (Sigma-Aldrich, St. Louis, MO, USA) in TBS at room temperature for 2 h to block non-specific binding, and then with primary antibody against MMP-2 or MMP-7 (1:100; Santa Cruz Biotechnology, Dallas, TX, USA, Cat. No. sc-10736 and sc-10737) in 1.5% BSA at 4 °C overnight. Slides were incubated with biotinylated secondary antibody (1:100; Vector Laboratories, Burlingame, CA, USA, Cat. No. BA-1000) for 2 h at room temperature, followed by incubation with an Avidin–Biotin–Peroxidase Staining kit (Thermo Scientific, Rockford, IL, USA). Detection was performed by using peroxidase substrate DAB (3.3-diaminobenzidine) kit (Thermo Scientific, Rockford, IL, USA). Hematoxylin (Sigma-Aldrich, St. Louis, MO, USA) was used as a counterstain. For negative controls, the primary antibody was omitted. Negative controls were run in parallel in all experiments. Images were captured by bright field microscopy under identical conditions of magnification and illumination.

### 2.8. MMP-2 and MMP-7 Activity Assay

The MMP-2 and MMP-7 activity was quantified in cell lysates and in the media by using QuickZyme Human MMP-2 activity assay v2 or QuickZyme human MMP-7 activity assay (QuickZyme Biosciences, Park, Leiden, The Netherlands) according to the manufacturer’s instruction.

### 2.9. Statistical Analysis

Results are presented as mean ± standard error of the mean (SEM). Mean differences were analyzed using Student’s *t*-test for single comparison or one-way analysis of variance followed by the Tukey’s multiple comparisons test for multiple comparisons when appropriate. Statistical analyses were performed using GraphPad Prism 7.02 software (GraphPad Software, San Diego, CA, USA). Values of *p* < 0.05 were considered significant.

## 3. Results

### 3.1. MMP-2 and MMP-7 Expression in LAM/TSC Cells

Considering that LAM/TSC cells do not express tuberin for a TSC2 germline mutation on exon 21 and an epigenetic modification, MMP-2 and MMP-7 expression were evaluated following incubation with the chromatin remodeling agent 5-azacytidine (5-Aza; 1 µM) that inhibits CpG DNA methylation causing tuberin expression, as previously shown [13]. 5-Aza induced tuberin expression in LAM/TSC cells with a very high efficiency in 96 h without altering the proliferation of LAM/TSC cells (Appendix A). In this condition, MMP-2 expression was greatly reduced in LAM/TSC tuberin-expressing cells compared to control LAM/TSC cells. On the contrary, MMP-7 levels were similar in the two populations, suggesting that MMP-7 expression is independent of tuberin expression (Figure 1A,B). It has been previously demonstrated that LAM/TSC cells spontaneously grow in adherent (90% of the total number of cells) and nonadherent status (10%), and that nonadherent TSC2^−/−^ cells invade a substrate by secreting MMP-7 [13,32]. We analyzed MMP-2 and MMP-7 expression in adherent and nonadherent LAM/TSC cell subpopulations, compared to the global MMP-2 and MMP-7 expression of all cells in a dish (that represents a mix of adherent and nonadherent cells). By Western blot analysis, both MMP-2 and MMP-7 expression were higher in nonadherent LAM/TSC cells compared to adherent cells, confirming that MMP-2 and MMP-7 have a role in the degradation of ECM protein that leads to the ability to switch from the adherent to the nonadherent status (Figure 1C,D). The MMP-2 and MMP-7 activity was comparable in adherent and nonadherent cells, and it was inhibited by the incubation with 5-azacytidine (Figure 1E,F).

### 3.2. Modulation of MMP-2 and MMP-7 Expression with Drugs

We previously demonstrated that LAM/TSC cell survival and proliferation strongly depends on EGF, since the treatment with anti-EGFR antibody causes cell death. For this reason, MMP-2 and MMP-7 levels in LAM/TSC cells were evaluated following 48 h of incubation with anti-EGFR antibody (5 µg/mL). To understand if the lack of tuberin might influence MMP-2 and -7 expression, we also treated LAM/TSC cells with rapamycin (5 ng/mL, 48 h) as a known drug targeting mTOR. The blockade of EGFR with anti-EGFR antibody and the inhibition of mTOR with rapamycin did not significantly alter MMP-2 and MMP-7 expression, even though MMP-7 appeared to be slightly affected by rapamycin (Figure 2A,B). mTORC1 phosphorylates the ribosomal S6 kinases S6 kinase on Thr^389^, which phosphorylates downstream substrates, such as ribosomal S6 [8]. Rapamycin leads to a rapid inhibition of S6K activity. As expected, phosphorylation of S6 was reduced by both drugs, with a stronger inhibition caused by rapamycin, and the proliferation of LAM/TSC cells was significantly reduced after 48 h exposure to both drugs (Appendix A). Interestingly, the active form of MMP-2 and MMP-7 was inhibited following incubation with anti-EGFR antibody and rapamycin (Figure 2E,F). To test the specific effect on MMPs activity, MMP-2 and MMP-7 expression were analyzed following incubation with doxycycline, a known MMPs inhibitor. Expression of both MMPs was reduced by incubation with 0.5 and 5 µM doxycycline for 48 h (Figure 2C,D), but S6 phosphorylation was not affected by doxycycline at the used concentrations. As expected, MMP-2 and MMP-7 active forms were significantly reduced by doxycycline at the concentration of 0.5 µM and were undetectable following the incubation with 5 µM doxycycline (Figure 2E,F). LAM/TSC cell proliferation was only slightly affected by doxycycline incubated for 48 h (Appendix A).

### 3.3. Secretion of MMP-2 and MMP-7 in LAM/TSC Cells

The treatment with the anti-EGFR antibody did not alter the secretion of MMP-2 and MMP-7 (Figure 3A,B). On the contrary, rapamycin partially inhibited MMP-2 secretion, even if the result was not statistically significant, as previously showed by Lee et al. [22]. Interestingly, rapamycin treatment greatly reduced MMP-7 secretion (Figure 3B).

We used MMP-2 and MMP-7 activity assays that allow the quantitative determination of MMPs in the active forms or pro-forms to evaluate in more detail the effects of the pharmacological treatments. Rapamycin inhibited the active form of MMP-2, which was unaltered by the anti-EGFR antibody (Figure 3C). The anti-EGFR antibody and rapamycin reduced the MMP-2 pro-form (Figure 3D). The MMP-7 active form was unchanged following the treatment with rapamycin, while the anti-EGFR antibody increased the active form of MMP-7 (Figure 3E,F). Following both treatments, the pro-forms of MMP-7 were undetectable. In order to assess if MMP-2 and MMP-7 secretion might be linked to the lack of tuberin expression, we incubated LAM/TSC cells with 5-Aza for 96 h. In this condition, MMP release was only slightly reduced (Figure 3A,B), unlike the significant inhibition of MMP-2 expression observed following 5-Aza treatment (Figure 1A). Similarly, the active form of MMP-2 and MMP-7 were only slightly inhibited by the incubation with 5-azacytidine, while both pro-forms were increased (Figure 3A,D–F).

As expected, doxycycline (0.5 and 5 µM) significantly affected MMP-2 and MMP-7 secretion and activity, with MMP-7 that was undetectable following 5 µM doxycycline incubation. The pro-form of MMP-2 was significantly reduced by doxycycline and the pro-form of MMP-7 undetectable.

### 3.4. Influence of MMP-2 and MMP-7 Secretion on LAM/TSC Cell Migration

As MMPs play a crucial role in cell migration, we investigated the migratory rate of LAM/TSC cells by wound-healing assay. Following the scratch in a confluent monolayer of control LAM/TSC cells, the wound was completely healed in 11 h (Figure 4).

The migration rate of LAM/TSC cells is impaired by the induction of tuberin expression, since the complete closure of the wound is obtained after 19 h in LAM/TSC cells treated with 5-azacytidine for 96 h. Moreover, the treatment with the anti-EGFR antibody and with rapamycin, incubated for 48 h before the scratch, significantly slowed the rate of LAM/TSC cell migration, with a complete closure of the wound at 25 h (Figure 4A). These results indicate that tuberin expression regulates the motility in LAM/TSC cells. Indeed, at the time point 11 h, when control LAM/TSC cells closed the wound, the opening of the wound for LAM/TSC cells treated with anti EGFR antibody, rapamycin, and 5-azacytidine was of 40%, 35%, and 42%, respectively (Figure 4B). As expected, the inhibition of MMPs secretion by doxycycline (0.5 and 5 µM) greatly slowed the healing rate, since the wound completely closed at 32 h with a migration distance higher than 90% at 11 h. Interestingly, the migration of LAM/TSC cells is higher compared to other cancer cells. For example, MCF7 breast cancer cells showed a significant delay in wound closure (83% open wound area at 11 h) and closed in approximately 40 h (Figure 4B).

### 3.5. Secretion of MMP-2 and MMP-7 during LAM/TSC Cell Migration

Considering the inhibitory effect of the drugs on the wound healing, MMP-2 and MMP-7 secretion was evaluated at the wound closure. MMP-2 secretion/hour was strongly decreased by anti-EGFR antibody and even more by rapamycin consistent with the drug effect on slowed migration (Figure 5A). Following the 5-azacytidine-induction of tuberin expression, MMP-2 secretion was only slightly reduced during the wound healing. Doxycycline drastically decreased the release of MMP-2.

Surprisingly, anti-EGFR antibody increased MMP-7 secretion compared to LAM/TSC control cells (Figure 5B). MMP-7 release was undetectable following treatment with rapamycin, 5-azacytidine, and doxycycline. During wound healing, the active forms of MMP-2 and MMP-7 were inhibited by anti-EGFR antibody and rapamycin, 5-azacytidine, and doxycycline (Figure 5C,E). Conversely, anti-EGFR antibody and rapamycin increased the pro-form of MMP-2 compared to control that was undetectable (Figure 5D,F). While anti-EGFR antibody increased the pro-form of MMP-7, rapamycin had an inhibitory effect similarly to doxycycline at the concentration of 0.5 µM. Following incubation with doxycycline at the 5 µM concentration, MMP-2 and MMP-7 active and pro-forms were undetectable. The induction of tuberin expression by 5-azacytidine strongly reduced the pro-form of MMP-2 and caused an increase of the pro-form of MMP-7. Even when measurable, the levels of the pro-forms of MMP-2 were very low compared to the active forms (Figure 5C,D).

Since CD147, also known as extracellular metalloproteinase inducer (EMMPRIN) and one of the activators of MMPs synthesis, can trigger matrix metalloproteinase inductions involved in ECM degradation, cell adhesion, and cell–cell interactions, we studied its expression following drug incubation. The percentage of LAM/TSC cells expressing CD147 was significantly reduced in LAM/TSC cells incubated for 96 h with 5-azacytidine (Figure 6). 5-azacytidine did not have any effect on CD147 expression in MCF-7 cells, suggesting a specific effect on the chromatin remodeling in the induction of tuberin expression of LAM/TSC cells. Anti-EGFR antibody and rapamycin incubation for 48 h significantly reduced the percentage of LAM/TSC cells expressing CD147 but at a slighter extent than 5-azacytidine.

### 3.6. Expression of MMP-2 and MMP7 in Lungs of a Mouse LAM Model

To study MMP-2 and MMP-7 expression and the effect of anti-EGFR and rapamycin in vivo, we took advantage of a mouse LAM model in which inhaled LAM/TSC cells in immunodeficient female athymic nude mice caused the formation of multiple lung nodules and enlarged alveolar spaces [16]. Thirty weeks after cell inhalation, MMP-2 was broadly expressed in lung nodules and localized in the epithelium as in control mice (Figure 7A).

Anti-EGFR antibody and rapamycin mainly reduced MMP-2 positivity in nodules but not in the epithelium (Figure 7A,B). This is consistent with the ability of both drugs to decrease the lung nodule dimensions [16]. Thirty weeks after cell inhalation in mice, MMP-7 was also detected in lung nodules and, in a similar way to control lungs, in epithelial airways (Figure 7B). Anti-EGFR antibody reduced MMP-7 expression in nodules and in epithelial cells, while rapamycin showed primarily a marked inhibition in nodules (Figure 7C,D).

## 4. Discussion

LAM is a progressive pulmonary disease characterized by the infiltration and proliferation of LAM cells in the lungs causing cystic destruction, and the formation of lung nodules and lymphatic vessels [19]. LAM cells exhibit the features and behaviors of a neoplasm, so this disease is now considered a low-grade, destructive, metastasizing neoplasm [10]. In cancer, proteolytic enzyme features of MMPs contribute to degrade ECM proteins, thereby facilitating tumor cell invasion, metastasis, and angiogenesis [1,2]. The upregulation of MMP-2 and MMP-9 in human LAM cells and in the serum of LAM patients, and the expression of MMP-1, -2, and -14 in LAM lung nodules indicate that MMPs contribute to LAM cell migration and lung destruction [18,19]. However, despite MMP characterization, their role in LAM and their involvement as therapeutic targets is not completely clear. Our study aims to deepen the involvement of MMP-2 and MMP-7 in LAM and TSC.

Consistent with previous in vitro studies, in LAM/TSC cells the absence of tuberin is associated with high expression and activity of MMP-2, as we observed a significant decrease in MMP-2 expression and in the levels of its active form following the incubation with the chromatin remodeling agent 5-azacytidine that causes tuberin expression [25,27]. Similarly, the activity of MMP-7 was dependent on tuberin expression, even if MMP-7 expression did not change. Consistent with this observation, the activity of MMP-2 evaluated in the media was reduced when tuberin was expressed, while the MMP-7 active form was unchanged. Very interestingly, in all the experimental groups, the pro-form of MMP-2 was detected at higher levels than the active form. Moreover, 5-azacytidine caused a slight increase of the pro-forms of MMP-2 and MMP-7 that may be caused by a non-specific inhibition of the DNA methylation of the chromatin remodeling agent. From these data, MMP-2 appears to be strongly related to tuberin expression in a more definite way than MMP-7. However, high levels of MMP-7 have been described in several types of invasive cancers, and the role of MMP-7 in mediating invasive capability in a dependent β-catenin way was reported in nonadherent Eker rat Tsc2^−/−^ cells. [32]. In the human LAM/TSC cells, MMP-2 and MMP-7 expression was also dependent on cellular adhesion while their activity did not appear to be related to the adhesion condition. Thus, multiple factors, including tuberin expression and cell adhesion, concur to control the expression, secretion, and activity of MMP-2 and MMP-7. We recently reported that MMP-2 and MMP-7 are higher in serum of LAM/TSC patients with minimal pulmonary disease that might be associated with a more active disorder and to LAM cell migration [20]. However, while elevated MMP-2 serum levels are related to pulmonary disfunction as they are observed in LAM and LAM/TSC patient serum, MMP-7 is higher only in the LAM/TSC population compared to sporadic LAM or TSC patients [20].

In LAM/TSC cells, the absence of tuberin was associated to high levels of CD147, an inducer of MMPs. Indeed, it has been reported that CD147 colocalized with MMP-2 and MMP-9 in LAM nodules and is overexpressed in bronchoalveolar lavage fluids [33]. However, the mechanism that controls MMPs synthesis is not unique, making it difficult to correlate CD147 only to tuberin expression. In fact, CD147 is highly expressed in several other pathological events such as interstitial pneumonia, failing myocardium, cirrhotic cell liver, cancer cell surface, and in lungs of chronic smokers [4,5]. CD147 induces MMP-1, -2, -3, and -9 in cancer cells and neighboring stromal cells, resulting in the regulation of ECM remodeling during inflammatory response and wound healing [34]. As MMP-related alterations of the ECM contribute to cell migration, invasion, and metastasis in cancer, a similar mechanism may facilitate LAM cell migration and entrance into the circulation. LAM/TSC cell migration was inhibited when LAM/TSC cells were induced to express tuberin. In an in vitro TSC2 model, it has been previously demonstrated that the absence of tuberin is responsible for cellular attachment and migration, which are modulated by α1β1 integrin [25]. In this study, adhesion and motility are independent from mTORC1. Conversely, in LAM/TSC cells, rapamycin reduced the migration and increased the time to heal the wound with a reduction of MMP-2 and MMP-7 activity and secretion. Except during cell motility, rapamycin had an inhibitory effect on the MMP-2 activity, without causing alteration in MMP-2 secretion and expression as previously described also in angiomyolipoma lacking-TSC2 cells and mouse embryonic cells that do not express Tsc1 or Tsc2 [22]. However, rapamycin strongly reduced MMP-7 secretion and activity in LAM/TSC cells without any effect on MMP-7 expression. LAM/TSC cell migration was reduced also by blocking the EGF receptor. As for rapamycin, anti-EGFR antibody in LAM/TSC cells inhibited MMP-2 secretion and activity only during wound healing but did not affect MMP-7 secretion whose significance is under investigation. Despite the different effect of rapamycin and anti-EGFR antibody on MMP-2 and MMP-7 secretion, both drugs had an inhibitory effect on the MMPs inducer CD147. A relationship between EGFR signaling and CD147 has been described in corneal epithelial cells in which EGF up-regulated CD147 to control proliferation and migration [35]. Likewise, in cutaneous squamous cell carcinoma, the inhibition of CD147, by using a chimeric anti-CD147 monoclonal antibody or a small interfering RNA against CD147, causes the reduction of EGFR [36].

Consistently with the in vitro data, MMP-2 and MMP-7 were highly expressed in lung nodules of the LAM model that we previously developed by injecting LAM/TSC cells in an athymic mouse. In another murine model, Gonchorova and colleagues demonstrated that MMP-7, -9, and -12 are significantly increased in TSC2 null lesions, while MMP-8 is enhanced in both TSC2-null and TSC2-expressing lesions, indicating that the expression of the different MMPs may depend on various factors, including tuberin expression [37]. Consistently, only the combination of rapamycin with simvastatin reduced MMP overexpression in TSC2-null growths, indicating that MMP expression only partially depends on mTOR activity [37]. Furthermore, in Tsc2-null myometrial tumors of a Tsc2 knockout mouse, MMP-2, -9, and -3 mRNAs were overexpressed and sensitive to rapamycin, which, however, had a poor effect on protein expression and activity that were dependent on estrogen expression [38]. In our LAM model, the expression of MMP-2 and MMP-7 in lung nodules was reduced by the inhibition of the two targets, mTOR and EGFR. Likely, the observed effects may be caused also by the decrease of nodule number and dimension.

Doxycycline showed an inhibitory effect on MMP-2 and MMP-7 secretion and activity and reduced the migratory ability of LAM/TSC cells. Doxycycline is a broad-spectrum antibiotic that has several properties such as antioxidant and anti-inflammatory effects that are likely independent of inhibition of MMP activity, which is achieved by the direct binding to the zinc ion domain in the catalytic site. Doxycycline was shown to decrease MMP expression in in vitro and in vivo models, such as in the lungs from cigarette smoke-exposed mice [39]. In addition to the MMP inhibitory effect, doxycycline has the capability to modulate gene expression through tetracycline-controlled transcriptional regulation [39]. Given the different effects and targets, with our data we cannot define if the inhibitory action of doxycycline in LAM/TSC cell migration might be due to the reduced MMP expression and/or secretion, and the decreased MMP activity in the presence of doxycycline might be caused either by an activity inhibition or by the reduced MMPs levels. The results of clinical studies with doxycycline in LAM treatment have shown conflicting results. Indeed, doxycycline treatment for 12 months reduced both urinary MMP-9 and serum MMP-2 levels in LAM patients, but FEV1 (forced expiratory volume in the 1st second) was stabilized or increased only in patients with mild spirometric abnormalities but not in patients with advanced disease [26]. In addition, in sporadic LAM or in LAM/TSC patients treated with doxycycline and followed for up 2 years, Chang and colleagues did not observe any significant effect on FEV1 [27].

In conclusion, the role of MMPs in LAM and TSC appears to be an underlying complex mechanism influenced by several factors. In LAM/TSC cells, MMP-2 and MMP-7 expression, secretion, and activity are dependent on tuberin expression, cellular adhesion, and migration. Rapamycin and anti-EGFR antibody inhibited the expression of the MMP inducer CD147; however, while MMP-2 secretion was affected by both drugs only during cell migration, MMP-7 levels were reduced by rapamycin in any condition but not by the anti-EGFR antibody, suggesting a different sensitivity to the two signaling pathways. However, in the LAM mouse model, both drugs caused a decrease of the MMP-2 and MMP-7 expression in lung nodules, likely related to their inhibitory effect on the MMP activity, mainly observed during wound healing. Considering our data, the unclear beneficial effect of doxycycline in LAM treatment may be caused by the complex and variable role of MMPs in the pathogenetic mechanism of LAM sustaining the need to better study the involvement of MMPs in the disease to evaluate a possible supportive therapeutic approach with doxycycline and to use MMPs as biomarkers for LAM.

## Figures and Tables

**Figure 1 biomedicines-09-01760-f001:**
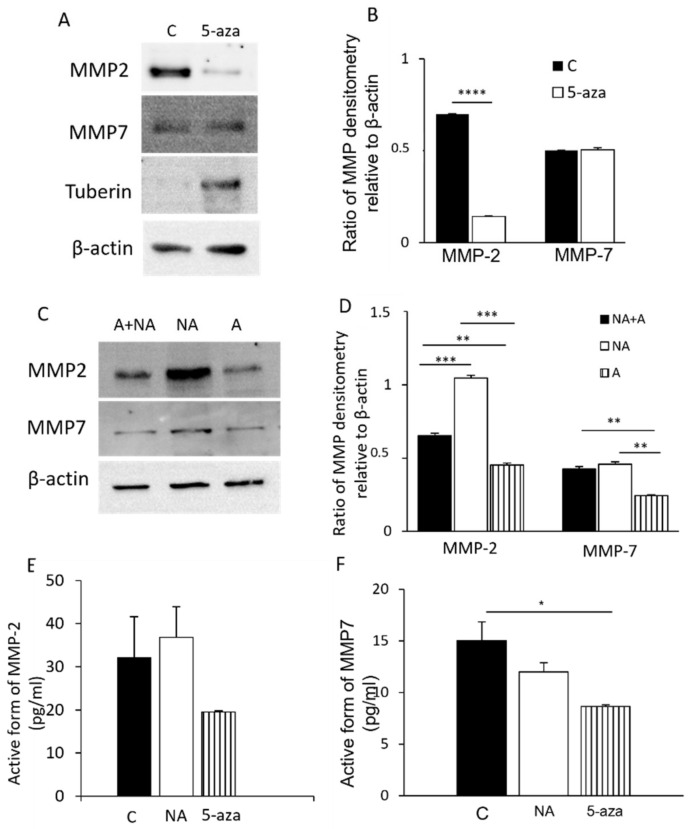
Tuberin expression and adhesion control of MMP-2 and MMP-7 levels in LAM/TSC cells: (**A**) Representative images of Western blot for MMP-2 and MMP-7 following incubation of 5-azacytidine (5-aza-1µM) for 96 h to induce tuberin expression. Tuberin expression was tested. β-actin was used as an internal reference of total protein. (**B**) Relative intensity by densitometric analysis was evaluated relatively to β-actin levels. The data are presented as mean ± SEM, *n* = 3 experiments. **** *p* < 0.0001 compared to C (Student T-test). (**C**) MMP-2 and MMP-7 levels were analyzed in adherent (**A**), nonadherent (NA), and N + NA conditions, and representative images are shown. β-actin was used as an internal reference of total protein. (**D**) Relative intensity by densitometric analysis was evaluated relatively to β-actin levels. (**E**) MMP-2 and (**F**) MMP-7 activity was analyzed by using a QuickZyme assay in control cells (**C**), NA cells, and in cells treated with 5-azacytidine (*n* = 3 experiments). For (**D**–**F**) statistical analysis was performed with ANOVA with Tukey test ± SEM (*n* = 3 experiments) * *p* < 0.05; ** *p* < 0.01; *** *p* < 0.001; **** *p* < 0.0001.

**Figure 2 biomedicines-09-01760-f002:**
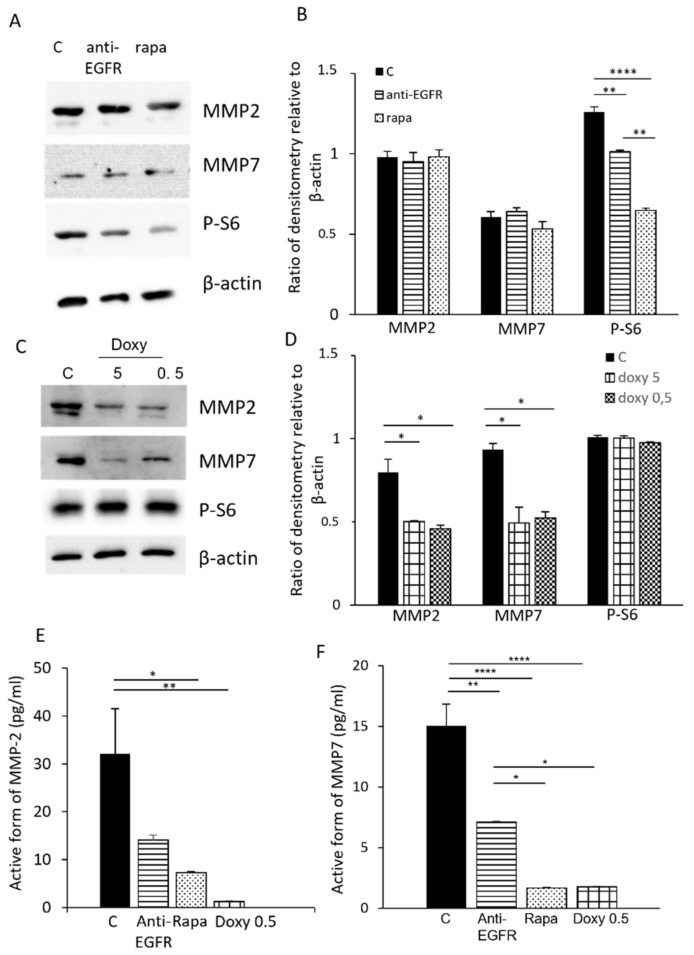
Effect of anti-EGFR antibody and rapamycin on MMP-2 and MMP-7 expression and activity. (**A**) MMP-2 and MMP-7 expression and S6 phosphorylation were analyzed by Western blot after incubation with anti-EGFR antibody (5 µg/mL) or rapamycin (5 ng/mL) for 48 h. Representative images are shown. (**B**) Relative intensity by densitometric analysis was evaluated relative to β-actin levels. (**C**) MMP-2 and MMP-7 expression, and S6 phosphorylation were evaluated by Western blot following incubation with doxycycline (5 and 0.5µM) incubated for 48 h. β-actin was used as an internal reference of total protein. (**D**) Relative intensity by densitometric analysis was evaluated relative to β-actin levels. (**E**) MMP-2 activity and (**F**) MMP-7 were analyzed by using a QuickZyme assay in control cells (**C**) and in cells incubated with anti-EGFR antibody, rapamycin, or doxycycline. ANOVA with Tukey test ± SEM (*n* = 3 experiments) * *p* < 0.05 ** *p* < 0.01; **** *p* < 0.0001.

**Figure 3 biomedicines-09-01760-f003:**
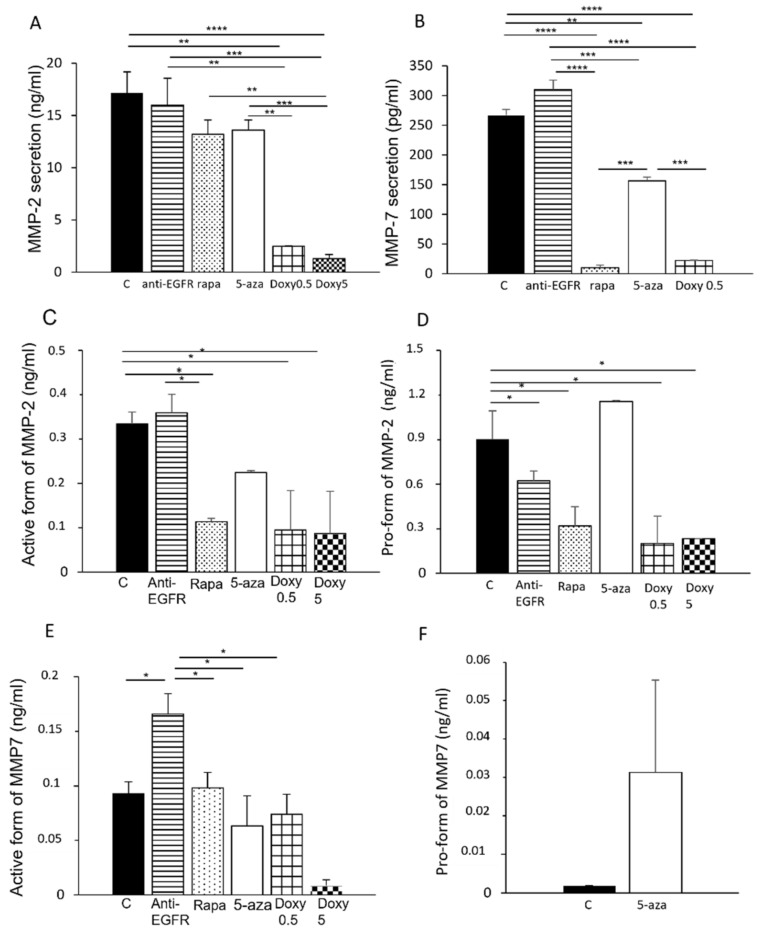
MMP-2 and MMP-7 secretion following treatment with anti-EGFR antibody and rapamycin and after tuberin expression. MMP-2 (**A**) and MMP-7 (**B**) secretion were evaluated by ELISA assay in control LAM/TSC cells and after incubation, for 48 h with anti-EGFR antibody, rapamycin, and doxycycline, and for 96 h with 5-azacytidine. The active form (**C**) or the pro-form (**D**) of MMP-2 was evaluated by using a QuickZyme assay in the culture medium of control LAM/TSC cells (**C**) and in cells incubated with anti-EGFR antibody, rapamycin, doxycycline (0.5 µM and 5 µM), and 5-azacytidine. The active form (**E**) or the pro-form (**F**) of MMP-7 was analyzed with a QuickZyme assay in the same experimental groups as in (**E**). Data represent means ± SEM of three independent experiments. ANOVA with Tukey test ± SEM (*n* = 3 experiments) * *p* < 0.05; ** *p* < 0.01 *** *p* < 0.001; **** *p* < 0.0001.

**Figure 4 biomedicines-09-01760-f004:**
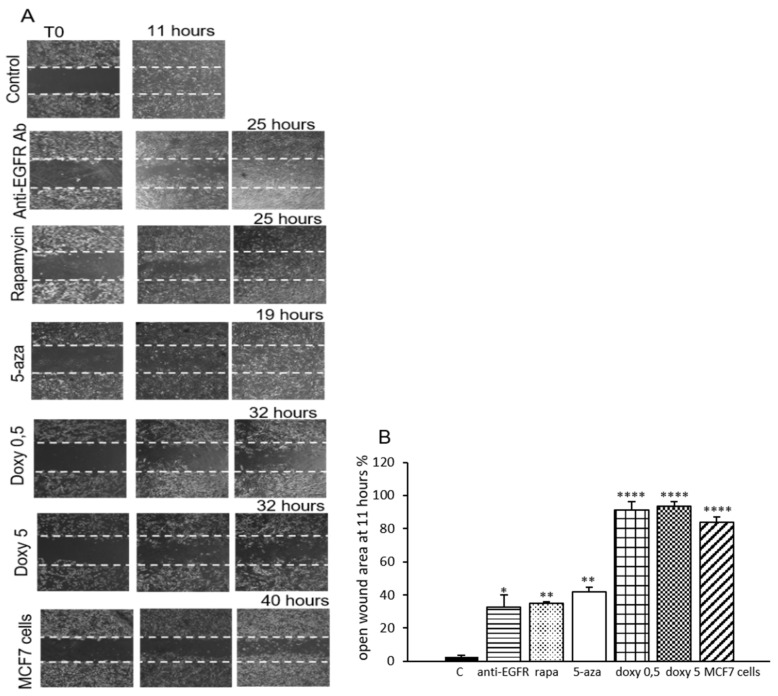
Migratory capability of LAM/TSC cells related to anti-EGFR antibody, rapamycin, and 5-azacytidine. (**A**) The figure shows representative images of LAM/TSC cell migration in a wound healing assay, following incubation with anti-EGFR antibody (anti-EGFR Ab; 5 µg/mL for 48 h), rapamycin (rapa; 5 ng/mL for 48 h), doxycycline (0.5 and 5 µM for 48 h), and 5-azacytidine (5-aza; 1 µM for 96 h) compared with migration of MCF7 cells. The lines of each panel indicate the initial wound (0 h; left panels of each set: T0). Phase contrast microscopy (magnification 10×). (**B**) Cell migration during wound healing was quantified 11 h after the wound by measuring the distance in pixels using ImageJ software. ANOVA with Tukey test ± SEM (*n* = 3 experiments) * *p* < 0.05; ** *p* < 0.01; **** *p* < 0.0001.

**Figure 5 biomedicines-09-01760-f005:**
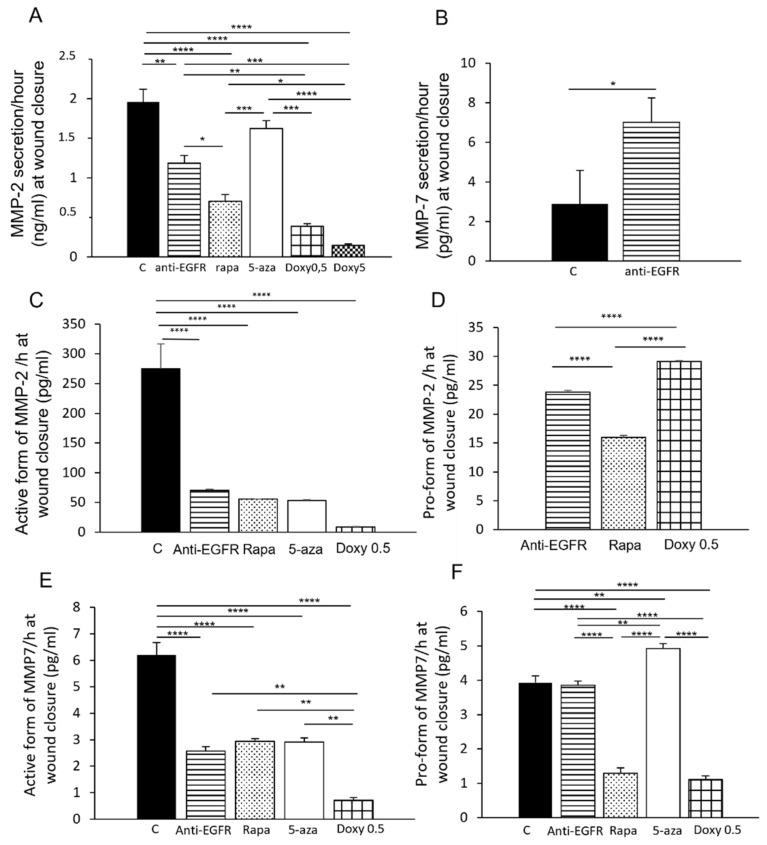
Effect of drug treatment on MMP2 and MMP-7 secretion during wound healing. MMP-2 (**A**) and MMP-7 (**B**) levels were analyzed by ELISA assay at wound closure in all experimental groups, as in Figure 4, and expressed as secretion/hour (ng/mL). The presence of the active form (**C**) or the pro-form (**D**) of MMP-2 in the culture medium of all the experimental groups, as in Figure 4, was evaluated by QuickZyme assay and expressed as secretion/hour (pg/mL). The presence of the active form (**E**) or the pro-form (**F**) of MMP-7 in the culture medium of all the experimental groups, as in Figure 4, was evaluated by QuickZyme assay and expressed as secretion/hour (pg/mL). ANOVA with Tukey test ± SEM (*n* = 3 experiments) * *p* < 0.05; ** *p* < 0.01; *** *p* < 0.001; **** *p* < 0.0001.

**Figure 6 biomedicines-09-01760-f006:**
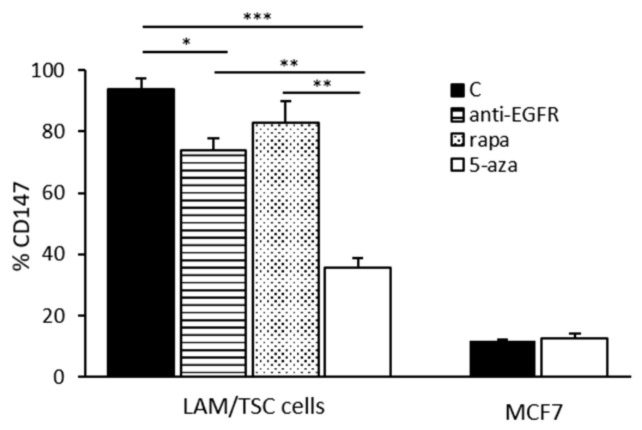
Effect of anti-EGFR antibody, rapamycin, and 5-azacytidine on CD147 expression. The percentage of LAM/TSC cells positive to CD147 was evaluated by flow cytometry after incubation for 48 h with anti-EGFR antibody and rapamycin for 48 h. The percentage of CD147 positivity was analyzed in control LAM/TSC cells and MCF7 cells, and after exposure to 5-azacytidine for 96 h. ANOVA with Tukey test ± SEM (*n* = 3 experiments) * *p* < 0.05; ** *p* < 0.01; *** *p* < 0.001. MCF7 c vs. MCF7 aza were analyzed by Student *t*-test.

**Figure 7 biomedicines-09-01760-f007:**
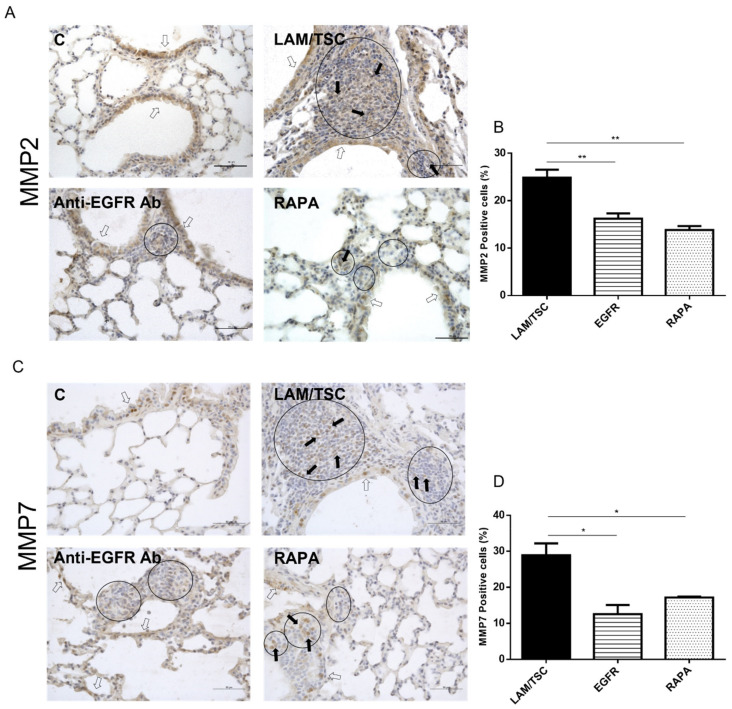
Expression of MMP-2 and MMP-7 in lungs of a LAM mouse model. Representative images of MMP-2 (**A**) and MMP-7 (**C**) expression in lung nodules of a LAM mouse model developed by LAM/TSC cell inhalation. Anti-EGFR antibody and rapamycin cause a reduction of MMP-2 and MMP-7 expression, and a decrease of nodule size. Nuclei were labelled with hematossilin (blue). Circles highlight nodules. Black arrows indicate MMP2 or MMP7 positive cells in nodules; white arrows indicate MMP2 or MMP7 positive cells in lung epithelium. Scale bars: 50 µm. Quantification of MMP-2 (**B**) or MMP-7 (**D**) positive cells in nodules. The percentage of positive cells on total cells is normalized on the numbers of nodules counted in each section. ANOVA with Tukey test ± SEM (*n* = 3 experiments) * *p* < 0.05, ** *p* < 0.01.

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
