# Peer review of "Differential Modulation of Matrix Metalloproteinases-2 and -7 in LAM/TSC Cells"

_biomedicines, 2021, doi:10.3390/biomedicines9121760_

Round 1

Reviewer 1 Report

  1. Abstract should contain a general sentence summarizing the main content of the study.
  2.  The list of abbreviations should be included at the end of the manuscript.
  3. Discussion is relatively short and should be extended.

Author Response

RESPONSE TO REVIEWER # 1 COMMENTS

  1. Abstract should contain a general sentence summarizing the main content of the study.

Response 1. As requested by the reviewer, the abstract has been reviewed

  1. The list of abbreviations should be included at the end of the manuscript.

Response 2. A list of abbreviations has been added at the end of the manuscript.

  1. Discussion is relatively short and should be extended.

Response 3. As suggested by the reviewer the discussion has been extended

Reviewer 2 Report

The manuscript by Dr. Ancona et al., evaluated the importance of MMP-2 and MMP-7 in LAM/TSC cells responsible for LAM cancer, and their modulation by different molecules. The article is well structured, easy to follow and well written. However, I think that there are just some major issues (and few minor ones) that should be addressed.

Major issues

1) The authors used cell lysates to perform the western blot (as said on line140). Although it might be useful as a proof that the MMPs produced are no longer in the cell, to evaluate the expression of MMPs a real time PCR seems more appropriate. In fact, the protein product of MMP-2 and MMP-7 are secreted in the medium and what is left inside the cell might not represent a real effect of the treatment, or it can highlight that the proteins are stuck in internal organelles or bound to plasma membrane. Thus, as you already did in other experiments (ELISAs), the election sample should be the culture medium. Therefore, if you want to present the data from expression, I strongly recommend to repeat the experiment presented in figure 1 and 2 by performing a real time PCR on purified mRNA from cells.

2) It would be nice to have some sort of histogram displaying the results from Immunohistochemistry presented in Figure 7, and arrows pointing to what is described in order to improve the readability of the Figure.

3) Although doxycycline is a broad-spectrum inhibitor of MMPs, I think that from their results the authors should distinguish and discuss between two distinct aspects of the molecule. a) Doxycycline treatment decreases the expression of MMPs (although as said on point 1, the results should be replicated through real time PCR to be bullet proof), which is not dependent by its action on enzymatic activity. b) Doxycycline treatment impairs the wound healing due to both decrease in MMPs amount and perhaps their activity (I think that these two effects are difficult to distinguish one from another). I think that the authors should discuss more in detail these aspects also in the discussion.

4) Since the active forms of the enzymes are the one truly responsible for the migration and the possible metastatic potential of the cells, it would be nice to see if there is some kind of change in the active forms of MMP-2 and MMP-7, perhaps with gelatin zymography (for MMP-2) and casein zymography (for MMP-7). Alternatively, they could use some kind of activity assay, which might be less challenging than zymography (cells need to be serum starved for a period because it interferes with the detection) although more expensive.

5) Although the authors cite the supplementary material, I was not able to evaluate it because it was not present in the submission.

Minor issues

1) in the methods section, please report also the catalog number for antibodies and important “uncommon” reagents (e.g. rapamycin, 5-azacytidine and so on). There is no need to add the catalog number of reagents like EDTA, deoxycholic acid and so on.

2) on figure 1 panels B and D, I think that there is an error in the y-axis title. In panel B it is written “Ratio of MMP-2 densitometry relative to beta-actin” and in panel D ““Ratio of MMP-7 densitometry relative to beta-actin”. Since both panels present each MMP (-2 and -7), I think that the authors would say “Ratio of MMP densitometry relative to beta-actin”.

3) On figure 2, panel A, there is no label explaining which western blot is MMP-2, MMP-7 and so on.

Author Response

RESPONSE TO REVIEWER # 2 COMMENTS

  • The authors used cell lysates to perform the western blot (as said on line140). Although it might be useful as a proof that the MMPs produced are no longer in the cell, to evaluate the expression of MMPs a real time PCR seems more appropriate. In fact, the protein product of MMP-2 and MMP-7 are secreted in the medium and what is left inside the cell might not represent a real effect of the treatment, or it can highlight that the proteins are stuck in internal organelles or bound to plasma membrane. Thus, as you already did in other experiments (ELISAs), the election sample should be the culture medium. Therefore, if you want to present the data from expression, I strongly recommend to repeat the experiment presented in figure 1 and 2 by performing a real time PCR on purified mRNA from cells.

Response 1. We thank the reviewer for the comment and the suggestion. However, considering that one of the regulators of the protein synthesis is mRNA degradation and not all the mRNA contents get translated for the various control steps (including experimental treatments), MMPs expression assayed by western blot and Elisa is the real final effect of the treatments. We understand the reviewer’s concerns and to answer them we think that the new data on MMPs active forms and pro-forms in LAM/TSC cells and in the medium, together with the protein expression (western blot) in LAM/TSC cells and the secretion in the media (ELISA), will give a more  exhaustive picture of the behaviour of MMPs in all the tested conditions. We think mRNA does not give the real expression of MMPs and mRNA presence is not enough to confirm the expression of the protein. Moreover, regarding the chance that proteins are stuck in internal organelles or bound to plasma membrane, we are confident that our lysis buffer and the method of protein solubilization are efficacious in breaking the protein interactions.

  • It would be nice to have some sort of histogram displaying the results from Immunohistochemistry presented in Figure 7, and arrows pointing to what is described in order to improve the readability of the Figure.

Response 2. Figure 7 has been improved by adding circles that highlight the nodules, and black and white arrows that indicate MMP-2 and -7 positive cells in nodules and in lung epithelium, respectively. Moreover, we quantified the MMP-2 and MMP-7 positive cells in the nodules that are expressed as percentage by counting the positive cells on total cells and normalizing this on the numbers of the nodules counted in each section. The normalization is necessary since the number of nodules differs between groups. We hope that Figure 7 is now easier to understand.

  • Although doxycycline is a broad-spectrum inhibitor of MMPs, I think that from their results the authors should distinguish and discuss between two distinct aspects of the molecule. a) Doxycycline treatment decreases the expression of MMPs (although as said on point 1, the results should be replicated through real time PCR to be bullet proof), which is not dependent by its action on enzymatic activity. b) Doxycycline treatment impairs the wound healing due to both decrease in MMPs amount and perhaps their activity (I think that these two effects are difficult to distinguish one from another). I think that the authors should discuss more in detail these aspects also in the discussion.

Response 3. We thank the reviewer for the comments. a) we agree that the doxycycline effect on MMPs expression is not dependent or is not necessarily dependent on enzymatic activity. We think that the data on MMPs active may help to, at least in part, answer to the question. We considered the issue in the discussion (line 607)  b) we agree with the reviewer that the action of doxycycline in wound healing might be due to MMPs amount of protein and MMPs activity. We added this point in the discussion (line 607)

  • Since the active forms of the enzymes are the one truly responsible for the migration and the possible metastatic potential of the cells, it would be nice to see if there is some kind of change in the active forms of MMP-2 and MMP-7, perhaps with gelatin zymography (for MMP-2) and casein zymography (for MMP-7). Alternatively, they could use some kind of activity assay, which might be less challenging than zymography (cells need to be serum starved for a period because it interferes with the detection) although more expensive.

Response 4. We thank the reviewer for the observation and following the request we analysed MMP-2 and MMP-7 activity. Given our difficulty to perform the experiments with gelatin zymography and casein zymography and to avoid the introduction of an experimental variable caused by the need with these techniques to starve the cells, we took advantage of QuickZyme human MMP-2 and MMP-7 activity assays that enable us to have a quantitative determination of the active MMP-2 and MMP-7 and the pro-forms following activation on the plate with APMA. The assay kit is described in the method section. The results are shown in Figure 5C, 5D, 5E, and 5F.

  • Although the authors cite the supplementary material, I was not able to evaluate it because it was not present in the submission.

Response 4. We submitted the Supplemental Material. We apologize if the prior submission was not correctly done.

Minor issues

  • in the methods section, please report also the catalog number for antibodies and important “uncommon” reagents (e.g. rapamycin, 5-azacytidine and so on). There is no need to add the catalog number of reagents like EDTA, deoxycholic acid and so on.

Response 1. The catalogue numbers were added as requested in the method section that has been updated.

  • on figure 1 panels B and D, I think that there is an error in the y-axis title. In panel B it is written “Ratio of MMP-2 densitometry relative to beta-actin” and in panel D ““Ratio of MMP-7 densitometry relative to beta-actin”. Since both panels present each MMP (-2 and -7), I think that the authors would say “Ratio of MMP densitometry relative to beta-actin”.

Response 2. The y-axis title has been corrected

  • On figure 2, panel A, there is no label explaining which western blot is MMP-2, MMP-7 and so on.

Response 3. We apologize for the mistake. The labels are added.

Reviewer 3 Report

  1. Please choose a different blot for Figure 1A. You have three experiments to choose from.
  2. Please include labels on Figure 2A. The blot quality looks poor (2A). Please add another blot from another experiment.
  3. Line 275: Please add background information on S6 phosphorylation
  4. The involvement of the respective MMPs in LAM could be better represented by performing activity assays in this paper. Gelatin zymography must be performed for MMP 2 and 9 in the respective experiments where their expression and secretion is higher. For example: Test activity of the respective MMPs in conditions tested in Fig 1A, 1C, 2C, 3 and 4. Appropriate MMP 7 activity assay must also be performed to test its activity.

Author Response

RESPONSE TO REVIEWER # 3 COMMENTS

  • Please choose a different blot for Figure 1A. You have three experiments to choose from.

Response 1. As requested by the reviewer we changed the blot for Figure 1A

  • Please include labels on Figure 2A. The blot quality looks poor (2A). Please add another blot from another experiment.

Response 2. The labels on Figure 2A have been added and the blots replaced.

  • Line 275: Please add background information on S6 phosphorylation

Response 3. As requested, background information regarding S6 phosphorylation have been added (Now line 320-322)

  • The involvement of the respective MMPs in LAM could be better represented by performing activity assays in this paper. Gelatin zymography must be performed for MMP 2 and 9 in the respective experiments where their expression and secretion is higher. For example: Test activity of the respective MMPs in conditions tested in Fig 1A, 1C, 2C, 3 and 4. Appropriate MMP 7 activity assay must also be performed to test its activity.

Response 4. We thank the reviewer for the suggestion. We performed experiments to evaluate MMP-2 and MMP-7 activity with appropriate activity assays. We used QuickZyme human MMP-2 and MMP-7 activity assays that allow the quantitative determination of MMP-2 and MMP-7 in the active or in the pro-form (activated by APMA). Zymography assay and gelatin assay need the cell starvation so we selected these assay kit to analyse MMPs activity without introducing any experimental variable. The results for LAM/TSC cells are shown in Figure 1E, 2 E and 1 F and 2 F for MMP-2 and MMP-7, respectively. Active form and pro-form evaluated in medium are shown in Figure 3C and 3D for MMP-2, and in Figure 3E and 3F for MMP-7. The MMP-s activities during the wound healing are shown in Figure 5C, 5D, 5E, and 5F.

Reviewer 4 Report

MINOR SPELL CHECK

Author Response

RESPONSE TO REVIEWER # 4 COMMENTS

MINOR SPELL CHECK

Spell check has been done

Round 2

Reviewer 2 Report

I feel that the authors replied to all my concerns.  

Reviewer 3 Report

Thanks for addressing my comments/concerns.